# Calmodulin as Ca^2+^-Dependent Interactor of FTO Dioxygenase

**DOI:** 10.3390/ijms221910869

**Published:** 2021-10-08

**Authors:** Michał Marcinkowski, Tomaš Pilžys, Damian Garbicz, Jan Piwowarski, Kaja Przygońska, Maria Winiewska-Szajewska, Karolina Ferenc, Oleksandr Skorobogatov, Jarosław Poznański, Elżbieta Grzesiuk

**Affiliations:** 1Institute of Biochemistry and Biophysics, Polish Academy of Sciences, Pawińskiego 5a, 02-106 Warsaw, Poland; mmarcinkowski@ibb.waw.pl (M.M.); tpilzys@ibb.waw.pl (T.P.); dgarbicz@ibb.waw.pl (D.G.); piwo@ibb.waw.pl (J.P.); kaja.przygonska@gmail.com (K.P.); mwin@ibb.waw.pl (M.W.-S.); skorobogatov.alx@gmail.com (O.S.); 2Center of Translational Medicine, Warsaw University of Life Sciences, Nowoursynowska 100, 02-797 Warsaw, Poland; karolina_ferenc@o2.pl

**Keywords:** FTO, calmodulin, calcium, MST, HDX

## Abstract

FTO is an *N*^6^-methyladenosine demethylase removing methyl groups from nucleic acids. Several studies indicate the creation of FTO complexes with other proteins. Here, we looked for regulatory proteins recognizing parts of the FTO dioxygenase region. In the Calmodulin (CaM) Target Database, we found the FTO C-domain potentially binding CaM, and we proved this finding experimentally. The interaction was Ca^2+^-dependent but independent on FTO phosphorylation. We found that FTO–CaM interaction essentially influences calcium-binding loops in CaM, indicating the presence of two peptide populations—exchanging as CaM alone and differently, suggesting that only one part of CaM interacts with FTO, and the other one reminds free. The modeling of FTO–CaM interaction showed its stable structure when the half of the CaM molecule saturated with Ca^2+^ interacts with the FTO C-domain, whereas the other part is disconnected. The presented data indicate calmodulin as a new FTO interactor and support engagement of the FTO protein in calcium signaling pathways.

## 1. Introduction

The FTO (FaT mass and obesity-associated) protein is an alpha-ketoglutarate and iron-dependent dioxygenase [1]. This two-domain protein is responsible for the removal of methyl groups from certain types of RNA [2]. FTO has been shown to be able to demethylate *N*^3^-methyluridine from RNA oligomers [3], *N*^6^-methyladenosine (*N*^6^-meA) from mRNA and snRNA, *N*^1^-methyladenosine from tRNA [4], *N*^6^,2-O-dimethyladenosine from both snRNA [4] and, the most efficiently, from the mRNA cap [5]. Importantly, the substrate specificity of FTO may be dependent on its localization within the cell [4]. The cap demethylation takes place in the cytoplasm only, snRNA demethylation occurs exclusively in the nucleus while the removal of the methyl group from mRNA happens regardless of its location [4].

The cellular FTO localization depends on its interaction with other proteins. It has been shown that FTO molecules actively move between the nucleus and the cytoplasm. The phosphorylation state of FTO in the position T150, introduced by casein kinase II, determines FTO presence in a particular cell compartment [6]. Phosphorylation of T150 maintains cytoplasmic location of FTO, while non-phosphorylated molecules are directed to the nucleus. Exportin-2 (XPO2) is one of the proteins involved in FTO transport through recognition of the NLS sequence present at the start of the FTO N-domain [7].

In the last 10 years, a number of other interactions influencing FTO parameters and regulating its functions in the cell have been identified. Lin and colleagues [8] have shown that FTO can interact with three (α, β, and γ) out of the four isoforms of calcium/calmodulin-dependent protein kinase II (CaMKII) [9], the members of the serine/threonine kinase family. CaMKII kinases phosphorylate a wide range of different substrates influencing processes such as cell development, proliferation, cellular transport, neuronal function and regulation of liver glucose [10,11,12,13,14,15]. For example, the interaction between FTO and CaMKII isoforms decreased the level of cAMP response element-binding protein (CREB) phosphorylation in human neuroblastoma cells (SK-N-SH). Next, the phosphorylation status of CREB protein determined the expression of specific genes associated with regulatory elements CRE [16]. The observed phosphorylation increased the expression of Neuropeptide Y receptor type 1 (NPY1R) and brain-derived neurotrophic factor (BDNF) proteins, which are also related to appetite and energy homeostasis.

Another protein with confirmed FTO interaction is the splicing factor, proline- and glutamine-rich (SFPQ). This nuclear protein [17] has been classified as a transcriptional factor participating in several metabolic pathways, e.g., in RNA transport, apoptosis or RNA repair. Together with Non-POU domain-containing octamer-binding proteins (p54nrb) and Paraspeckle component 1 (PSPC1), it forms a multifunctional family of *Drosophila* behavior/human splicing (DBHS) proteins with a tendency to create homo- and heterodimers [17]. Song and colleagues [18] noted that there are many physiological pathways and pathological conditions with participation of FTO and SFPQ proteins. They found that SFPQ, the RNA-binding protein, interacts with the FTO C-domain [18] and, when RNA was inspected, both proteins were close to each other on the chain. Moreover, SFPQ protein recognizes the CUGUG sequence and promotes FTO-directed *N*^6^-meA demethylation close to the indicated sequence. It is likely that the regulation of *N*^6^-meA in physiological and pathological states depends on SFPQ-FTO interaction. Additionally, FTO interaction with methionyl-tRNA synthetase (MRS) links the protein with energy metabolism [19]. MRS is responsible for connection methionine with the corresponding *t*-RNA; thus, MRS-FTO interaction suggests FTO involvement in amino acid metabolism.

Phosphorylation of certain serines in FTO by Glycogen synthase kinase 3 (GSK-3) affects the lifetime of the protein by its further ubiquitination, resulting in proteolytic decomposition [20]. On the other hand, Tai and colleagues [21] have shown that threonine phosphorylation in FTO gives the opposite effect.

The above-mentioned reports suggest that FTO appears in a network of interactions with other cellular elements. Further, our analysis of protein lysates from head and neck cancer (HNSCC), with the use of size exclusion chromatography indicated that FTO was located in a wide spectrum of molecular fractions—from several tens to several hundred kDa. This confirmed the presence of a wide range of the possible FTO interactors and determined the goal of our present study. Initially, we checked whether potential FTO interactors involve proteins that are known to recognize specific amino acid sequences. The Calmodulin Target Database [22] enabled us to identify sequences from several species indicating that FTO C-domain potentially interacts with calmodulin (CaM), an important signaling protein. It consists of two globular lobes, N- and C-, each possessing two calcium-binding sites connected through a flexible linker [23]. We investigated the FTO–CaM relationship, because calmodulin is one of the major regulators of Ca^2+^-dependent signaling pathways in all eukaryotic cells. Its high conservation during evolution, broad spectrum of functions via interactions with several dozen of other proteins and the fact that its presence is crucial for many tested organisms [24] underscore the importance of the research into the interplay between CaM and its partners.

In this study, we used modern biophysical and biochemical methods (MicroScale Thermophoresis, hydrogen–deuterium exchange) to precisely describe, for the first time, FTO–CaM protein–protein interactions and demonstrate FTO calcium dependency. This may suggest that FTO is involved in calcium signaling pathways.

## 2. Results

### 2.1. FTO Protein Forms Homo- and Heterocomplexes In Vivo

We have already shown that overexpression of the FTO protein occurs in head and neck cancer (HNSCC) and that FTO level was positively correlated with the tumor size, indicating the involvement of this protein in cancer metabolism [25]. This raises the question whether FTO interactions with various proteins, already reported for CaMKII [8], GSK-3 [20] or SFPQ [18], explain its engagement in the metabolic processes other than cancerogenesis, such as adipogenesis [4], osteogenesis [26] or neural development [27] in normal cells. To answer the question of whether regulation of FTOs occurs at the gene or protein level, it must be first determined if in vivo FTO exists as monomer, dimer or in protein complexes of higher molecular mass.

Here, size exclusion chromatography (SEC) was performed for the purified recombinant FTO obtained from the baculovirus expression system (^BES^FTO), as well as for samples from the HNSCC tissues and specific cell lines, both cancerous (HeLa and U87) and non-cancerous (HEK293). In the absence of other proteins, FTO (58 kDa) showed the ability to form dimers [28]. This form predominates in vitro outside the cell. There were no higher oligomeric forms of the protein, even at high concentrations of ^BES^FTO, that were clearly visible in the Coommasie stained SDS-PAGE gel (Figure 1a). On the other hand, FTO derived from cellular lysates behaved differently. Western blot analysis of fractions eluted from the column showed that FTO appeared in almost all the fractions, suggesting that in cancer tissues and cancer cell lines the protein exists in highly diverse forms, ranging from monomers up to large complexes (Figure 1b–j). Interestingly, in human embryonic kidney cells (HEK293), under the tested conditions, FTO appeared rather as a mixture of dimers and predominating monomers. This behavior strongly suggests that FTO may interact with other proteins, which is clearly evidenced by changes in the elution profile. Moreover, the data obtained in the HEK293 cell line suggest that other proteins or small compounds may interfere with FTO homodimerization.

### 2.2. The Manner of FTO and CaM Interaction

Bearing in mind the elution profiles (Figure 1) suggesting interaction with other proteins, we looked at FTO sequences signatures determining interactions with other proteins. Inspection of the Calmodulin Interactors Database (http://calcium.uhnres.utoronto.ca/ctdb; accessed on 2 April 2021) showed that the C-terminal domain of FTO is likely to interact with CaM (Figure 2a). Analysis of ten FTO sequences from distinct species pointed out two regions putatively involved in this interaction: the first one corresponds to residues 375–395 of human FTO sequence and is common for *Danio rerio, Xenopus laevis, Gallus gallus, Mus musculus* and *Rattus norvegicus*; the second one, covering residues 430–455, is common for *Rattus norvegicus, Canis familiaris, Pongo pygmaeus abelii* and *Sus scrofa*. Taking into consideration that CaM is an important signaling molecule, we decided to investigate this interaction experimentally.

We use MicroScale Thermophoresis (MST) to follow FTO–CaM interaction. The fluorescently labeled FTO sample was titrated with unlabeled bovine testicle calmodulin (^BT^CaM) in the presence of FTO cofactors (Fe^2+^ and 2-OG) and Ca^2+^. FTO from both expression systems displayed one binding site (Figure 2b,c), suggesting that the protein may form a heterodimer with CaM (FTO–CaM), with the affinity of the first one at the nanomolar level (*K*_D_ = 26 ± 10 nM for ^EC^FTO and *K*_D_ = 34 ± 18 nM for ^BES^FTO). To confirm the interaction between FTO and CaM, we also used another type of fluorescent tag directed to surface-exposed lysine residues. However, with this approach, labeling of ^BES^FTO perturbed the FTO–CaM interaction (Appendix A; for details, see Materials and Methods). To study the interaction at the nanomolar level, we changed the labeling strategy and titrated fluorescently labeled ^BES^CaM with unlabeled FTO (in the presence of cofactors and Ca^2+^). In this case, *K*_D_ was 9 ± 1 nM and 6 ± 1 nM for ^EC^FTO and ^BES^FTO, respectively (Figure 2d,e). The dissociation constants determined for FTO from both expression systems were similar.

### 2.3. Both CaM Lobes Show the Ability to Interact with FTO

To identify the protein regions taking a part in the FTO–CaM interaction, we used the hydrogen–deuterium exchange (HDX) technique. Taking into consideration that there were no substantial differences in binding affinity of ^EC^FTO and ^BES^FTO to calmodulin, we tested recombinant proteins ^BES^FTO and ^BES^CaM. By investigating the ^BES^CaM HDX profiles, we found that the regions responsible for Ca^2+^ binding tended to show a lower rate of exchange relative to residues located in the central part. This is consistent with the supplier’s application note for holo calmodulin [29]. This effect was evident for the three detectable calcium-binding sites (Figure 3). While the addition of the ^BES^FTO did not alter this relationship, in its presence changes in the accessibility of almost all of the ^BES^CaM regions were as early as after 60 s of exposure to D_2_O, with a strong increase in exchange rate for the three observed calcium-binding loops: residues 21–32, 94–105 and 130–141 (Figure 3). The effect of the ^BES^FTO presence was also observed for the second calcium-binding loop in a second experiment, in which peptides covering the previously mentioned sequence were identified (Appendix A). In the reverse experiment, ^BES^CaM only slightly destabilized the FTO structure, mostly in the C-terminal domain, with the effect becoming noticeable after 1 h of incubation (Appendix A). These results again confirm FTO–CaM interaction. Further analyses were conducted to obtain more insight into FTO–CaM complex formation.

Detailed analysis of ^BES^CaM HDX data pointed to the specific behavior of particular peptides (Figure 4a) belonging to the regions responsible for calcium binding. These peptides were observed simultaneously in the both lobes of the ^BES^CaM and they seem to be present in two populations of ions (Figure 4b,c): one with the profile similar to free ^BES^CaM, while the other displaying an increased exchange rate. This effect is detectable at two ^BES^CaM and ^BES^FTO molar ratios: 1 to 1.5 (Figure 4a) and 1 to 2 (Appendix A), clearly suggesting that ^BES^CaM–FTO interaction affects only one lobe of the calmodulin (either N or C terminal), with the other one remaining unchanged, but never both simultaneously. Moreover, there is no specific preference for which of the two lobes is affected more by ^BES^FTO presence.

### 2.4. FTO–CaM Interaction Is Ca^2+^-Dependent

Keeping in mind that the interaction of CaM with many proteins is Ca^2+^-dependent, we studied the effect of Ca^2+^ on the interaction with FTO, using the MST technique for ^EC^FTO labeled on the HisTag. As mentioned above, in the presence of the Ca^2+^, the stronger *K*_D_ for the FTO–CaM complex was 26 ± 10 nM (Figure 5, blue; Figure 3a). However, in the absence of Ca^2+^, FTO–CaM interaction was hardly detectable and only one binding event with *K*_D_ ~17 µM was observed (Figure 5 orange).

### 2.5. FTO Interacts with CaM via C-Terminal Domain

A model of the FTO–CaM complex based on the obtained results is presented on Figure 6a. The N-terminal pair of EF-hand motifs of CaM was found to interact efficiently with FTO, while the C-terminal was oriented unfavorably relative to FTO. In order to avoid the steric hindrances between the C-terminal CaM domain and FTO, various conformations of the CaM interdomain linker were tested, while keeping coordinates of the N-terminal pair of EF-hands fixed. The best model was obtained for the linker in a helical conformation, adopted from the holo form of CaM (1CLL), with the C-terminal domain of CaM remaining calcium free. This model is consistent with the experimental results indicating that the presence of calcium is crucial for interaction and that half of the calmodulin displays different properties than the other half.

After detailed investigation of CaM behavior in the presence of the FTO, we experimentally verified FTO C-domain interaction with calmodulin. First, the interaction between bovine testicle calmodulin (^BT^CaM) and FTO C-domain (hereafter called CD-FTO) expressed in the *E.* Coli system was tested by a pull-down assay where the presence of His-tagged CD-FTO delayed the removal of ^BT^CaM from nickel beads (Figure 6b). Next, complex formation between these proteins was monitored by gel filtration (Figure 6c). The CD-FTO (22.2 kDa) elution profile showed three clearly visible peaks corresponding to large aggregates (at 11.0 mL), a homodimer (at 13.8 mL), and monomer (at 14.9 mL). ^BT^CaM (17 kDa) was eluted as the singular peak at 14.5 mL. As indicated by its molecular mass, it should have eluted at 15.2 mL, similarly to the myoglobin, but because of its non-globular shape, ^BT^CaM was eluted earlier. After gel filtration of the CD-FTO–^BT^CaM mixture, the elution profile showed three distinct peaks. The first likely corresponds to large aggregates (at 11.0 mL), it was also present in the CD-FTO elution profile. The second (at 12.4 mL) may result from the initial stages of protein aggregation. The CD-FTO–^BT^CaM mixture was eluted mainly as a peak at 14.1 mL, with the location corresponding to a putative heterodimer. The fact that this peak eluted between the two latter peaks of pure CD-FTO (13.8 mL for the homodimer and 14.9 mL for the monomer) is consistent with the lower molecular mass of ^BT^CaM compared with CD-FTO (17 kDa vs. 22.2 kDa).

## 3. Discussion

The FTO demethylase is a member of the ALKBH family of proteins connected with both adipogenesis, osteogenesis, maintaining bone mass and heart regeneration, as well as civilization diseases such as obesity, type 2 diabetes, cancer and others [28]. Many studies have demonstrated that FTO is involved in crucial cell processes; however, not much is known regarding the regulation of its activity. Here, we found that FTO–CaM protein–protein interaction plays a key role in understanding the role of FTO in cellular homeostasis. The amino acid sequence of FTO comprises a fairly conserved motif predicted to be recognized by CaM. Using several complementary approaches, we verified that FTO can indeed interact with CaM at nanomolar concentrations in the presence of Ca^2+^. Moreover, the interplay between FTO and CaM was only slightly affected by the type of the expression system used and only in the part of the experiments. Thus, the presence of posttranslational modifications, especially serine phosphorylation [28], occurring mostly in the FTO N-terminal domain did not significantly affect FTO–CaM binding. The obtained results showing nanomolar *K*_D_ are likely to be biologically relevant, because FTO in tissues occurs at nanomolar—or even lower—concentrations that can be detected by sensitive methods, such as Western blot [30]. On the other hand, total CaM concentration can be much higher. In certain states, it can reach several µM [31,32] or even 40 µM [33]. However, the free pool of CaM in a cell may be much lower due to the interactions of CaM with other protein targets, e.g., GAP-43 or RC3 [34,35]. However, it is important to note that the CaM/Ca^2+^ affinity to FTO is stronger than many other CaM interactors measured by similar biophysical methods. Here, we estimated *K*_D_ for CaM/Ca^2+^/FTO complexes to equal ca. 10–30 nM. In contrast, for the CaM/Ca^2+^/CaMKII-specific peptide complex (residue 290–309), the *K*_D_ measured by MST technique was much higher (190 nM). Meanwhile, the neurogranin protein binds the apo form of calmodulin with even lower affinity: *K*_D_ = 890 nM [36]. Finally, another newly identified CaM interactor, phosphoinositide-interacting regulator of transient receptor potential (PIRT), shows Ca^2+^-dependent affinity with *K*_D_ of 350 nM and 60 µM in the absence and presence of Ca^2+^, respectively [37].

Having confirmed FTO–CaM interaction, we then investigated the one binding site mode, in comparison with other known calmodulin interactors. According to the Calmodulin Target Database, 350 protein sequences have been shown to bind CaM. The process of CaM-targeted amino acid sequence binding is known to occur both in the presence, as well as in the absence, of Ca^2+^.

Earlier, Nissen and co-workers [38] analyzed a number of CaM-targeted complexes and stored them in the Protein Data Bank (PDB). Unsurprisingly, the presence of four Ca^2+^ within the CaM-targeted complex was found in the majority of studied PDB structures, because CaM is known to be able to interact with the target proteins/peptides upon binding four Ca^2+^. Multiple examples of four Ca^2+^-bound CaM-targeted complexes have been reported so far, predominantly featuring CaM-dependent protein kinases and phosphatases [39]. The two groups are characterized by a number of CaM-binding modes, namely 1–14, 1–5–10 and 1–10–16. However, CaM has also been shown to form Ca^2+^-free complexes. This includes, for example, the CaM-NaV1.5 (IQ-motif) interaction that occurs within the C-lobe of CaM and IQ-motif in α6 of NaV1.5 or the previously mentioned interactions with neurogranin and PITR. Notably, in these cases, the CaM C-lobe adopts a semi-open conformation [40]. Another Ca^2+^-free complex is CaM-Myosin-5A with the following topology: the C-lobes of the two CaM molecules bind to the N-terminals of the IQ motifs of NaV1.5 in an antiparallel orientation [41]. The last type of CaM-targeted complexes features partial CaM saturation with Ca^2+^. The CaM-binding domain (CaMBD) of small-conductance Ca^2+^-activated potassium channel, SK2-a, forms a complex with a single CaM molecule at each of the two ends of the SK-CaMBD dimer consisting of two helix-loop-helix motifs. Interestingly, only the N-lobes of CaM molecules bound Ca^2+^ wrapped around three α-helices of CaMBDs [42]. The ApoCaM-SK2-a binding mode is different: a single SK CaMBD binds to the C-lobe of CaM, resulting in 1:1 stoichiometry. A helical fragment of the CaMBD was shown to interact with one of the CaM C-lobes [43].

Here, our structural analysis showed that the interaction with FTO destabilized the CaM, especially the calcium-binding regions; nevertheless, the reciprocal effect was negligible. This may be explained by FTO homodimerization, when interfaces of the FTO–FTO homodimer and the FTO–CaM heterodimer are similar [28]. Possibly, both interactions require FTO to maintain the same particular structure. Further, FTO–CaM interaction on the nanomolar level was also Ca^2+^-dependent, with negligible *K*_D_ in the absence of Ca^2+^. This feature is typical for CaM interaction with numerous partners [44]. Importantly, this indicates that the proteins are likely to form a complex when the concentration of free Ca^2+^ is sufficiently high and that they may be regulated by calcium concentration. For example, such complexes may be present in neural cells where the CaM level influences synapse development [36] and FTO activity regulates neuronal development and the proliferation and differentiation of adult neural stem cells [27].

Finally, we modelled the structure of the newly discovered FTO–CaM complex. The generated structure indicates similarity between the FTO–CaM-binding mode and that of the previously mentioned SK2-a-CaM. Indeed, FTO–CaM interaction at nanomolar concentrations is Ca^2+^-driven, but only two Ca^2+^ binding to the single lobe of CaM participate in the complex formation, while the second lobe remain free of Ca^2+^ (Figure 6a). This model is also consistent with the HDX results. The predicted stoichiometry of the biological FTO–CaM complex is 1:1, with the CaM molecule wrapped around the FTO C-terminal domain. Importantly, this type of interaction suggests that CaM–FTO may mediate FTO interaction with other partners.

The discovery of this exclusive CaM–FTO interaction type sheds a new light on the results of previous FTO studies [8]. It is worth mentioning that both proteins interact with numerous other proteins, including three isoforms of CaMKII. Li and coworkers have shown that an increased FTO expression delays forskolin-mediated dephosphorylation of one of CaMKII’s cellular targets: cAMP response element-binding protein (CREB) in human neuroblastoma cells [8]. However, direct influence of FTO on CaMKII enzymatic activity as well as FTO phosphorylation by this kinase were not observed. Due to the fact that CaM is a CaMKII activator, our results show new connections between CaM, CaMKII and FTO phosphorylation by CaMKII. Non-physiological FTO concentration of 25 µM in the performed experiment and the finding that the affinity of CaM/Ca^2+^ to FTO seems to be higher than to CaMKII likely results in a shortage of free CaM necessary for CaMKII activation. Clearly, this relationship indicates that interplay between FTO, CaM and CaMKII needs further study [8]. Finally, it cannot be excluded that FTO–CaM interaction may affect the binding of other FTO interactors. For example, as previously mentioned, the FTO C-terminal domain interacts with SFPQ protein. This interaction affects substrate specificity of the FTO protein, thus the regulatory role of calmodulin has to be taken into consideration.

## 4. Materials and Methods

### 4.1. Gel Filtration Chromatography

For investigation of the molecular state of endogenous human FTO, fast protein liquid chromatography (FPLC) was performed. Recombinant FTO from baculovirus expression system (^BES^FTO) was expressed and purified as described previously [28]. Protein extracts from selected cancer tissues and HeLa, HEK293, and U87 cell lines were prepared as described previously [25]. A total of 900 µg of ^BES^FTO and 200 µg of each protein lysate sample of a final volume of 500 µL were loaded onto a SEC650 column (#780-1650, BIO-RAD, Hercules, CA, USA) equilibrated and eluted with the PBS buffer. The eluate was monitored with NGC system ChromLab™ (BIO-RAD, Hercules, CA, USA) at 215, 260, and 280 nm. Protein standards (#1511901, BIO-RAD, Hercules, CA, USA) for calibration were analyzed using the same method. Next, fractions of 1 mL were precipitated by the addition of 20% pyrogallol red-molybdate (PRM, #611A, Sigma-Aldrich, Darmstadt, Germany) solution, subsequently re-dissolved in Laemmli buffer and analyzed by Western blot as before [25].In the case of the CaM and FTO C-terminal domain interaction study, protein samples: 1.5 nmol of bovine testicle CaM (^BT^CaM, supply by Medicago Company; Uppsala, Sweden, 7.6 µM) and/or 3.0 nmol of recombinant His-tagged C-terminal ^EC^FTO domain (CD-FTO), (15.2. µM) were incubated for 10 min at 4 °C in SEC1 buffer of a final volume of 200 µL and loaded onto a SEC650 equilibrated and eluted with the SEC1 buffer. The eluate was monitored with NGC system ChromLab™ (BIO-RAD) at 215, 260 and 280 nm. Protein standards for calibration were analyzed by the same method.

### 4.2. In Silico Analysis of FTO Sequence

The amino acid sequence of human FTO was analyzed to identify possible protein interactors with the set of software: ProSite (https://prosite.expasy.org/; accessed on 1 April 2021), The Calmodulin Target Database (http://calcium.uhnres.utoronto.ca/ctdb; accessed on 2 April 2021), and MyHits (https://www.expasy.org/resources/myhits/; accessed on 3 April 2021). After detection of the region potentially binding calmodulin in C-terminal domain of human FTO, the amino acid sequence of FTO C-domain from selected species were investigated by The Calmodulin Target Databes. Next, the similarity between sequences was evaluated by the Clustal Omega software (https://www.ebi.ac.uk/Tools/msa/clustalo/; accessed on 4 April 2021).

### 4.3. cDNA Cloning and Plasmid Constructs

For use in the Baculovirus Expression System (BES), the coding sequence of CaM was PCR-amplified with suitable primers and inserted into the pENTRY-IBA5 donor vector between Xbal and HindIII sites. The insert was then moved to the pLSG-IBA35 destination vector (StarGate® cloning system, IBA Life Science, Göttingen, Germany).

To obtain the FTO C-terminal domain, the reaction was performed in a 50 µL of a final concentration of CON1. The cDNA was amplified in an Eppendorf Mastercycler^®^ according to the program: 95 °C for 2 min, followed by 30 cycles of 95 °C for 20 s, 61.5 °C for 30 s, 70 °C for 1 min and finally 70 °C for 10 min. The primers used for the FTO C-terminal domain construct generation are shown in Supplementary Table 1. Next, the PCR product was incorporated into empty pET28a(+) plasmid with the use of In-Fusion^®^ HD Cloning Plus (Takara Bio, #638909), according to protocol. Then, ~100 ng of plasmid was mixed with 200 µL of *E. coli* competent DH5α bacteria solution, incubated for 30 min on ice, heat shocked for 30 s at 42 °C, 800 µL of SOC buffer was added and the bacterial suspension was incubated for 1 h at 37 °C. The bacteria were plated on LB-agar plates and incubated overnight at 37 °C. Single colonies were cultured in liquid LB overnight at 37 °C. Bacteria were then harvested and plasmid was isolated with the use of GeneJET Plasmid Miniprep Kit (#K0503, Thermo Scientific™, Waltham, MA, USA).

All plasmids were verified by sequencing.

### 4.4. Expression and Purification of ^EC^FTO and ^BES^FTO

The procedure was performed as before [28].

### 4.5. Expression and Purification of ^BES^CaM in Baculovirus Expression System (BES)

The recombinant baculoviruses were generated directly in Sf21 insect cells by co-transfection with plasmid containing the human *CaM* coding sequence and FlashBacUltra virus DNA following the manufacturer’s protocol (FlashBAC™ system, Oxford Expression Technologies, Oxford, United Kingdom). Insect cells were infected with a recombinant baculovirus at a MOI = 4. After 72 h, the cells were harvested, resuspended in PRP4 lysis buffer and homogenized. After centrifugation (20,000× *g*, 10 min.), proteins His-tagged at the N-terminus were purified and verified by size exclusion and SDS-PAGE. The purified proteins were flash-frozen in liquid nitrogen and stored at −80 °C.

### 4.6. Expression and Purification of FTO C-Terminal Domain in E. coli

Plasmids harboring cDNA encoding C-terminally His-tagged C-terminal domain of human FTO, were introduced into *E. coli* BL-21. Bacteria were cultured at 37 °C to OD_600_ = 0.8 in 2 L of LB supplemented with kanamycin (50 µg/mL) and chloramphenicol (25 µg/mL), induced with 1 mM IPTG, cultured for 16 h at 16 °C, harvested and handled as described for full length ^EC^FTO [28].

### 4.7. Preparation of Protein Samples

Procedure was performed as before [28].

### 4.8. Pull-Down Assay

Protein samples: 67 pmol of ^BT^CaM (1.3 µM) and/or 130 pmol of CD-FTO (2.6 µM) were incubated 10 min at 4 °C in PDA1 buffer of final volume 50 µL. Next, 50 µL of Ni-charged resin suspension (Profinity™ IMAC Ni-Charged Resin, BIO-RAD, #156-0133) was added and incubated overnight at 4 °C with mixing. The resin was then washed with increasing concentrations of imidazole in the PDA1 buffer: five times with 10 mM imidazole (50 µL per wash), three times with 150 mM and one with 0.5 M. Subsequently, the samples were diluted by 4x SDS-PAGE loading buffer, boiled 5 min and separated on the Mini-PROTEAN TGX 4–15% gradient gels (Bio-Rad). Proteins were visualized by Coomassie staining.

### 4.9. Microscale Thermophoresis (MST)

Proteins were labeled with Nanotemper^®^ dye NT-647-NHS (lysine labeling) or NTA-RED-HisTag (HisTag labeling) following the manufacturer’s protocol, and their fluorescence was used to monitor the thermophoretic effect.

HisTag labeling: For each labeling, 125 pmol of protein (50 µL of 2.5 µM solution) and 125 pmol of the dye were used. After labeling, the protein was transferred to experimental buffer MST2 alone or supplemented with 0.5 mM CaCl_2_ (MST3).Lysine labeling: The procedure was performed as before [28]. Briefly, for each labeling, 1 nanomole of protein (50 µL of 20 µM solution) and 2.5 nanomoles of the dye were used. The labeling efficiency was determined according to the manufacturer’s protocol. Only samples with the labeling efficiency in the range of 65–90% were used in further experiments to make sure that in most cases only one molecule of the dye would be attached to the protein. After labeling, the protein was transferred to experimental buffer supplemented with 0.5 mM (NH_4_)_2_Fe(SO_4_)_2_, 0.5 mM CaCl_2_ and 1 mM 2-OG (MST1). For the case of ^BES^FTO labeling, the procedure perturbed the interaction: a ligand-induced fluorescence change occurred and only one weak and probably unspecific dissociation constant was visible, with a value of 24.7 ± 3.6 µM for the ^BES^FTO–^BES^CaM complex (Appendix A). This suggests that protein labeling may occur on lysines proximal to the FTO and CaM interface and such labeling strategy for this type of experiment should be avoided.

Pseudo-titration experiments were performed on a Monolith NT.115 RED/BLUE device equipped with holder allowing for parallel measurement of 16 samples placed in capillaries. Emission of the dye was monitored at 670 nm (excitation at 650 nm by LED laser) at room temperature. The LED laser power and MST power were optimized for each system individually, according to the labeling efficiency. A total of 2–3 repetitions of each experimental set-up with different MST powers were performed.

The data were analyzed using R 3.3.3 software [45] according to a standard two-state model describing a 1:1 equilibrium between the unbound and bound forms of the labeled protein [46], as well as using Origin software (www.originlab.com; accessed on 28 April 2021) according to the three-state model describing coexistence of 1:1 and 1:2 complexes [47].

### 4.10. Hydrogen–Deuterium Exchange (HDX)

Protein samples at two different ratios: 1–3 nmol of ^BES^FTO (30 µM) and/or 2 nmol ^BES^CaM (20 µM); 2–4 nmol of ^BES^FTO (40 µM) and/or 2 nmol ^BES^CaM (20 µM), were incubated for 30 min at 4 °C in an incubation buffer supplemented with 0.5 mM CaCl_2_ (HDX1). The samples were then diluted 10-fold with HDX2 reaction buffer (prepared in D_2_O). The hydrogen–deuterium exchange was terminated at: 0 s, 10 s, 1 min, 20 min, 1 h, and 24 h. The exchange was terminated by decreasing pH to approx. 2–3 with HDX3 stop buffer and the samples were immediately frozen in liquid nitrogen. The extent of hydrogen–deuterium exchange was determined using a nanoACQUITY UPLC system (Waters Corporation, Milford, MA, USA). The samples, after initial protein digestion on a trypsin column, were analyzed by liquid chromatography coupled with mass spectrometry (LC-MS). Each experiment was repeated four times. The data were analyzed with the DynamX HDX Data Analysis Software 3.0 (Waters Corporation, Milford, MA, USA).

### 4.11. Molecular Modeling

The structure of the FTO–CaM heterodimer was initially modeled based on the structure of CaM interacting with a fragment of small conductance potassium channel SK2-A (4qnh; A), whose CaM-interacting helix2 (D445-T486) was aligned iteratively with helix12 of FTO (P427-S458) using varying sets of successive residues located in the two helices. The best model was obtained by aligning residues ^476^QANTLVDLAKT^486^ of SK2-A with residues ^439^LASLTARQNLR^449^ of FTO.

The final model was obtained after several rounds of simulated annealing, in which the coordinates of FTO and the N-terminal domain of CaM together with the local fold of the C-terminal CaM domain were fixed, and the linker was kept in helical form.

## 5. Conclusions

The last 10 years of intensive research into the role of the FTO protein in metabolism have identified a number of transcripts as substrates of this demethylase, affecting cell metabolism and mediating FTO specificity. Our discovery of CaM as another FTO interactor supports the thesis that this demethylase is regulated by other proteins. We discovered the ability of FTO, especially its C-terminal domain, to interact in vitro with the single lobe of a CaM molecule in a Ca^2+^-dependent manner. Moreover, only one part of CaM interacts with FTO, whereas the other one remains free. This interaction may be a part of regulatory mechanisms that FTO is subjected to, due to the fact that other FTO interactors, such as FTO itself and SFPQ protein, also bind to the FTO C-domain. The recognition of the role of CaM in the FTO interaction helps broaden the understanding of the function of this RNA modification demethylase. This finding is important because the multiplicity of FTO interactors may explain the involvement of this protein in a number of different metabolic pathways.

## Figures and Tables

**Figure 1 ijms-22-10869-f001:**
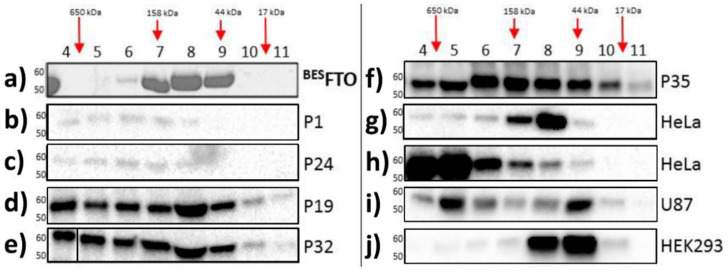
Analysis of cell lysates containing FTO with the use of gel filtration chromatography. Each panel shows FTO contents in particular fractions (4–11) after SEC of protein lysates. (**a**) recombinant ^BES^FTO exists as the mixture of the monomeric and dimeric forms; (**b**–**f**) HNSCC samples where FTO is present also in high molecular weight fractions, corresponding to masses up to 650 kDa; (**g**,**h**) HeLa cell samples where FTO is present in different fractions depending on the samples; (**i**) U87 cell samples where FTO exist simultaneously in low and high molecular weight fractions; (**j**) HEK293 samples where FTO monomeric forms are dominant. The results were visualized by Coomassie staining (**a**) or Western blot analysis (**b**–**j**). Antibodies used in experiment were verified previously [25].

**Figure 2 ijms-22-10869-f002:**
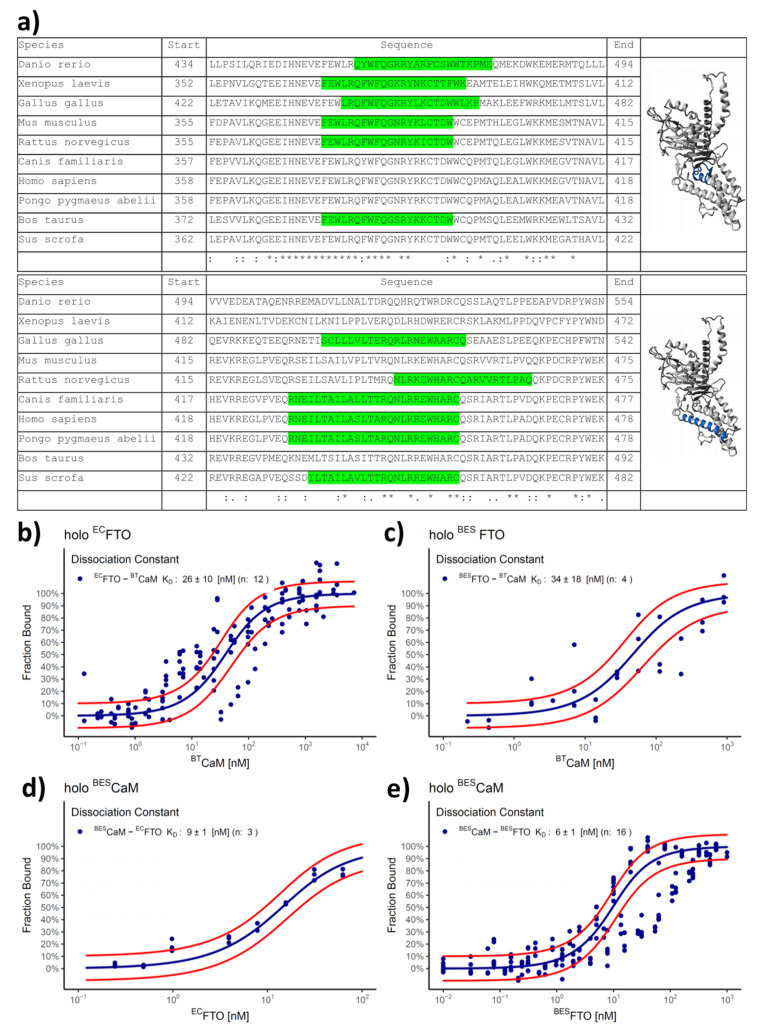
In silico and in vitro study of FTO–CaM interaction. (**a**) Predicted CaM interaction sites of selected species proposed by Calmodulin Target Database in FTO sequences. Only the sequences with a significant score (>7) are highlighted in green. Such regions are present mostly in FTO C-terminal domain. On the right, human FTO structure with marked sequences of interest (blue) presented on the protein surface. Annotations from Clustal Omega software: “*”—identical amino acids; “:”—high similarity between amino acids; “.”—low similarity between amino acids; “ ”—no identity. (**b**–**e**) FTO–CaM interaction in the solution containing 0.5 mM Fe^2+^, 1 mM 2-OG, and 0.5 mM Ca^2+^ analysed with the use of MST (**a**,**b**). Representative MST pseudo-titration data for the binding of ^BT^CaM to a HisTag-labeled ^EC^FTO (**a**) or ^BES^FTO (**b**). Both FTO preparations showed one visible transition, corresponding to the CaM-binding site to FTO, at the nanomolar level. (**d**,**e**) Titration of the unlabeled ^EC^FTO (**d**), ^BES^FTO (**e**) to a lysine-labeled ^BES^CaM. Both FTO preparations showed one visible transition, corresponding to the CaM-binding site to FTO, at the nanomolar level. Circles represent experimental data, blue lines follow the resulting model of the one binding site, and red lines represents 95% confidence limits for this model. *K*_D_ are shown as estimated value ± standard deviation. n—number of repetitions.

**Figure 3 ijms-22-10869-f003:**
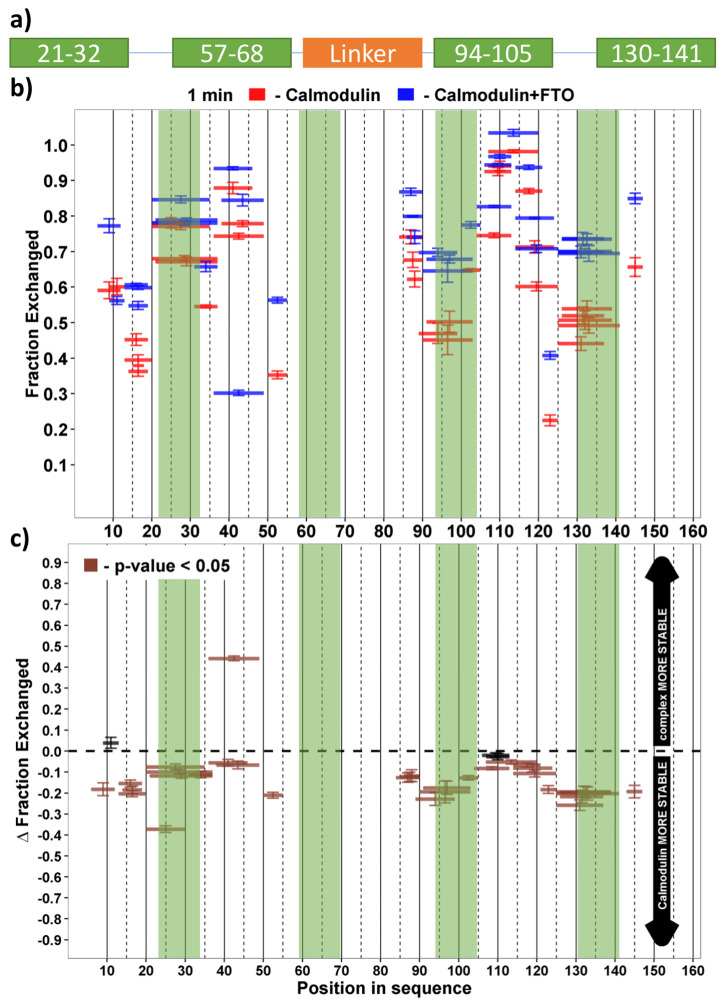
HDX analysis of the ^BES^FTO effect on the solvent accessibility of ^BES^CaM. (**a**) Representation of ^BES^CaM showing known regions: four EF-hand motifs responsible for calcium binding (green), an alpha helix linker which joins the N- and C- parts of calmodulin (orange). (**b**) HDX pattern of ^BES^CaM in complex with ^BES^FTO following a 60 s incubation with D_2_O. ^BES^CaM alone (red bars) and in the presence of the ^BES^FTO (blue bars). *x*-axis: position of peptides in amino acid sequence; *y*-axis: fraction of peptide that had undergone HD exchange. The mean of four experiments is shown. Error bars show both values measured. (**c**) Differences between deuteration of ^BES^CaM peptides alone and in the presence of the ^BES^FTO were derived by subtraction of exchange levels shown in (A). Brown bars indicate peptides for which the differences measured in repeated experiments satisfied the *t*-test with *p* < 0.05. Interaction of ^BES^CaM with ^BES^FTO affects most parts of the first protein, significantly decreasing the protein hydrogen–deuterium exchange, even after 1 min. Destabilization of ^BES^CaM caused by ^BES^FTO was the largest in the calcium-binding regions that cover residues 21–32, 94–105 and 130–141. Each experimental setup was repeated four times.

**Figure 4 ijms-22-10869-f004:**
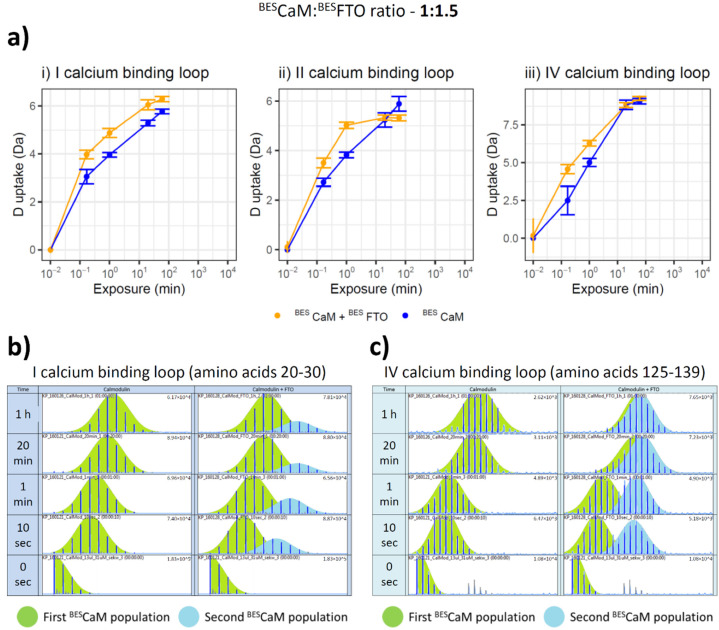
Effect of the ^BES^CaM-^BES^FTO interaction on the deuterium relative uptake of the selected peptides from ^BES^CaM calcium-binding loops. (**a**) HDX was monitored in the absence (blue) or presence (orange) of 30µM ^BES^FTO. Representative deuterium uptake plots for peptides (i) 20–30, (ii) 56–69 and (iii) 125–139. Each example shows that ^BES^FTO presence increases the HD exchange rate of the given peptides of ^BES^CaM. The deuterium incorporation was monitored at 10 sec, 1 min, 20 min and 1 h. Standard deviations for each time point are plotted as error bars. All measurements were performed in quadruplicate. (**b**,**c**) HDX-MS raw data of calmodulin peptides: (**b**) 20–30 and (**c**) 125–139, showing differential HDX between ^BES^CaM in the absence (left panel) and presence (right panel) of ^BES^FTO. After only 10 sec, the separation of two peptide populations, the basic (green) and the additional (blue), can be seen in ^BES^CaM incubated with ^BES^FTO, showing a higher exchange rate represented by an isotopic envelope made by peaks of higher m/z ratio.

**Figure 5 ijms-22-10869-f005:**
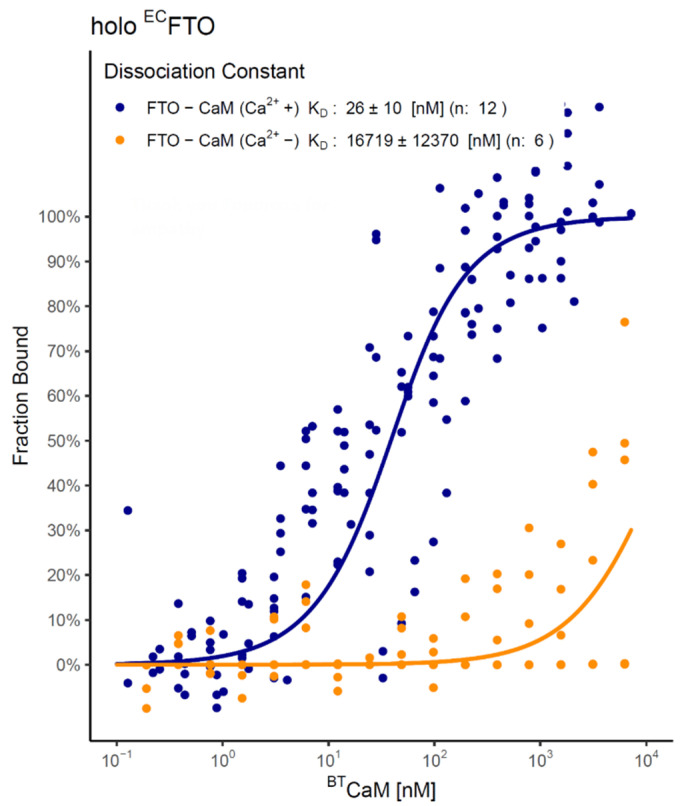
MST analysis of FTO–CaM interaction in a solution devoid of Ca^2+^ (orange) or supplemented with 0.5 mM Ca^2+^ (blue). Analysis of ^EC^FTO–^BT^CaM interaction in the presence of 0.5 mM Fe^2+^ and 1 mM 2-OG. Ca^2+^ absence increased *K*_D_ by 1000-fold, making interaction hardly detectable. The plot represents complex level of labeled protein at a given ^BT^CaM concentration for each separate sample (data points) and for modeled equilibrium between monomer and dimer (straight lines).

**Figure 6 ijms-22-10869-f006:**
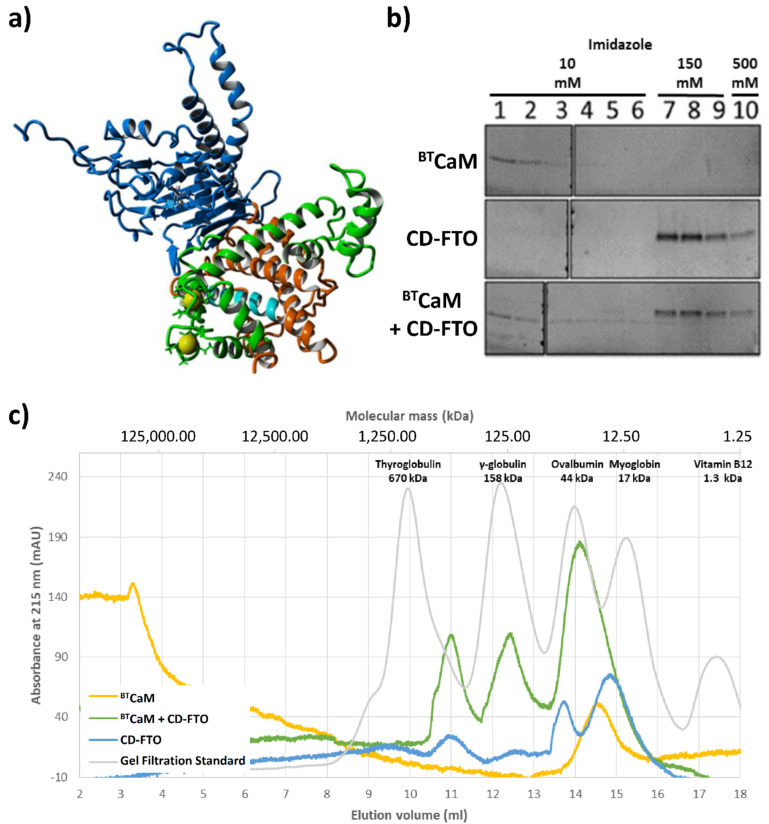
In silico and in vitro study of the interaction between CaM and C-domain of ^EC^FTO (CD–FTO). (**a**) The model of the FTO–CaM–Ca^2+^ complex. Two calcium (yellow)-binding loops of the N-terminal part of CaM (green) are close to the CaM–biding region (cyan) located in the C-terminal domain (orange) of the FTO (blue). (**b**,**c**) Interaction between the ^EC^FTO C–terminal domain with calmodulin. (**b**) Pull-down assay of ^BT^CaM with the CD–FTO. CD–FTO was used as bait, with selected concentrations of imidazole: 10 mM, 150 mM, 0.5 M. Numbers indicate specific elution fraction in the presence of Fe^2+^, Ca^2+^, 2–OG. The presence of the CD–FTO protein in the incubation mixture delayed the elution of ^BT^CaM. Elution experiments with ^BT^CaM only or just CD–FTO were performed as controls. ^BT^CaM alone did not interact with Ni^+^-resin. (**c**) Gel filtration chromatography of ^BT^CaM (yellow), CD–FTO (blue) and their mixture (green) samples in the presence of 0.5 mM Mn^2+^, 0.5 mM Ca^2+^ and 1 mM 2–OG. The CD–FTO chromatogram shows the presence of two fractions corresponding to monomeric and dimeric state, as well as large unspecific aggregates. ^BT^CaM is present only in one, low molecular mass form. The chromatography of the two protein mixtures reveals the presence of one significant peak corresponding to the expected mass of the CD–FTO–^BT^CaM complex.

**Table 1 ijms-22-10869-t001:** Compositions of buffers.

	Technique	Experiment	Buffer Composition
PBS	Gel filtration chromatography	Estimation of the molecular size of protein sample	10 mM Na_2_HPO_4_, 1.8 mM KH_2_PO_4_, pH 7.4, 137 mM NaCl, 2.7 mM KCl
SEC1	Gel filtration chromatography	Estimation of the molecular size of protein sample	50 mM MES, pH 6.5, 150 mM NaCl, 10% (*v*/*v*) glycerol_,_ 1 mM 2-OG, 0.5 mM MnCl_2_, 0.5 mM CaCl_2_
CON1	Construct preparation	Preparation of plasmids carrying C-terminal domain of FTO	1x buffer stock solution for KOD Hot Start DNA Polymerase, 1.5 mM MgCl_2_, 0.2 mM each dNTP, 0.5 µM each primer, 1 U of KOD Hot Start DNA Polymerase (Sigma-Aldrich, #71086), 1 ng of pET-28a(+) carrying hFTO cDNA
SOC	Bacteria preparation	Transformation	10 mM NaCl, 2.5 mM KCl, 2% (*w*/*v*) tryptone, 0.5% (*w*/*v*) yeast extract, 20 mM glucose, 10 mM MgCl_2_, 10 mM MgSO_4_
PRP1	Protein purification	Lysis of the bacteria	10 mM Na_2_HPO_4_, 1.8 mM KH_2_PO_4_, pH 7.4, 137 mM NaCl, 2.7 mM KCl, 10% (*v*/*v*) glycerol, 0.1% (*v*/*v*) TWEEN^®^ 20, 0.1% (*w*/*v*) lysozyme, 5 mM imidazole, 5 mM β-mercaptoethanol
PRP2	Protein purification	Elution from Ni-Charged Resin	10 mM Na_2_HPO_4_, 1.8 mM KH_2_PO_4_, pH 7.4, 137 mM NaCl, 2.7 mM KCl, 10% (*v*/*v*) glycerol, 0.1% (*v*/*v*) TWEEN^®^ 20, 150 mM imidazole, 5 mM β-mercaptoethanol
PRP3	Protein purification	Dialysis	50 mM Tris-HCl, pH 7.5, 150 mM NaCl, 50% (*v*/*v*) glycerol, 5mM β-mercaptoethanol
PRP4	Protein purification	Lysis of the BES Insect cells	50 mM HEPES, pH 7.5, 150 mM NaCl, 10% (*v*/*v*) glycerol, 5 mM β-mercaptoethanol
PDA1	Pull-down assay	Examination of protein–protein interaction	50 mM MES, pH 6.5, 150 mM NaCl, 10% glycerol (*v*/*v*), 2 mM L-ascorbic acid, 1 mM 2-OG, 0.5 mM (NH_4_)_2_Fe(SO_4_)_2_, 0.5 mM CaCl_2_
MST1	Microscale thermophoresis	Experimental buffer for ^EC^FTO and ^BES^FTO with the lysine labeling	50 mM MES pH 6.1, 150 mM NaCl, 0.05% (*v*/*v*) TWEEN^®^ 20, 2 mM L-ascorbic acid, 1 mM 2-OG, 0.5 mM (NH_4_)_2_Fe(SO_4_)_2_, 0.5 mM CaCl_2_
MST2	Microscale thermophoresis	Experimental buffer for ^EC^FTO and ^BES^FTO with the HisTag labeling	10 mM Na_2_HPO_4_, 1.8 mM KH_2_PO_4_, pH 7.4, 137 mM NaCl, 2.7 mM KCl, 10% (*v*/*v*) glycerol, 2 mM L-ascorbic acid, 0.5 mM (NH_4_)_2_Fe(SO_4_)_2_, 1 mM 2-OG
MST3	Microscale thermophoresis	Experimental buffer for ^EC^FTO and ^BES^FTO with the HisTag labeling	10 mM Na_2_HPO_4_, 1.8 mM KH_2_PO_4_, pH 7.4, 137 mM NaCl, 2.7 mM KCl, 10% (*v*/*v*) glycerol, 2 mM L-ascorbic acid, 1 mM 2-OG, 0.5 mM (NH_4_)_2_Fe(SO_4_)_2_, 0.5 mM CaCl_2_
HDX1	Hydrogen–deuterium exchange	Sample incubation	50 mM MES pH 6.1, 150 mM NaCl, 0.004% (*v*/*v*) TWEEN^®^ 20, 2 mM L-ascorbic acid, 1 mM 2-OG, 0.5 mM (NH_4_)_2_Fe(SO_4_)_2_, 0.5 mM CaCl_2_
HDX2	Hydrogen–deuterium exchange	HDX reaction	30 mM Tris-DCl, pD 7.5 (pH_READ_+0.4), 150 mM NaCl in D_2_O (#DLM-4DR-99.8-PK, Cambridge Isotope Laboratories, Inc.)
HDX3	Hydrogen–deuterium exchange	HDX stopping	2 M glycine, pH 2.4, 107 mM NaCl

## Data Availability

The data presented in this study are available on request from the corresponding authors.

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
