# Peer review of "Calmodulin as Ca2+-Dependent Interactor of FTO Dioxygenase"

_ijms, 2021, doi:10.3390/ijms221910869_

Round 1

Reviewer 1 Report

In the manuscript entitled “Calmodulin as calcium dependent interactor of FTO dioxygen-2ase”  the Authors show that FTO C-domain binds CaM and that the interaction is Ca2+ -dependent but independent of FTO phosphorylation. Also, the Authors found that only one part of CaM interacts with FTO, and suggest that the other one, most probably, interacts with other ligands. Generally, the presented results suggest that FTO is a molecule engaged in cellular calcium signaling pathway.

In general, manuscript presents a solid peace of work; experiments are properly designed and performed. An interesting issue, which could add more to this work, is confirmation of the interaction between FTO and CaM in the cell, for instance by PLA (proximity ligation assay) and identification of ligand(s) which bind to the FTO-CaM complex.

The manuscript is quite well written but English should be improved before the paper acceptance.

Minor points:

In Figure 1A not only monomer and dimer but also higher molecular weight species of FTO are visible, so it is not in agreement with the statement written in the Results section.

In section 2 “Results”, all subsections have the same numbers.

Line 254 – sentence is not finished.

“Ca2+“ means ions, so if you write “Ca2+“ you do not need to add word “ions”.

Regarding “calcium” you should decide and write “calcium ions” or “Ca2+” everywhere.  The same is for number of calcium ions, in some places there is “four” and in some just “4”. Also, instead of calcium dependent” should be “Ca2+-dependent”.

Author Response

Response to Reviewer 1 Comments

Point 1: In general, manuscript presents a solid peace of work; experiments are properly designed and performed. An interesting issue, which could add more to this work, is confirmation of the interaction between FTO and CaM in the cell, for instance by PLA (proximity ligation assay) and identification of ligand(s) which bind to the FTO-CaM complex.

  • Response 1: No doubt, the Reviewer is right that performing experiments to confirm the FTO-CaM interaction in the cell is necessary for understanding the metabolic role of this interaction. This is the focus of our further research, in which we will also use the technique suggested by the Reviewer.

Point 2: The manuscript is quite well written but English should be improved before the paper acceptance.

  • Response 2: English has been corrected by native speaker.

Point 3: In Figure 1A not only monomer and dimer but also higher molecular weight species of FTO are visible, so it is not in agreement with the statement written in the Results section.

  • Response 3: Thank you for this comment. The presence of FTO in fractions 6 and 7, corresponding to higher masses than the dimer and monomer, is due to the large amount of FTO sample used in the experiment. In the case of the size exclusion chromatography technique, the large amount of material results in earlier elution of the sample from the column. The lack of higher molecular forms than the dimer for purified recombinant FTO protein was demonstrated in our earlier work "Effect of Posttranslational Modifications on the Structure and Activity of FTO Demethylase" (doi.org/10.3390/ijms22094512).

Point 4: In section 2 “Results”, all subsections have the same numbers.

  • Response 4: It was corrected.

Point 5: Line 254 – sentence is not finished.

  • Response 5: It was corrected.

Point 6: “Ca2+“ means ions, so if you write “Ca2+“ you do not need to add word “ions”.

  • Response 6: It was corrected.

Point 7: Regarding “calcium” you should decide and write “calcium ions” or “Ca2+” everywhere.  The same is for number of calcium ions, in some places there is “four” and in some just “4”. Also, instead of calcium dependent” should be “Ca2+-dependent”.

  • Response 7: It was corrected.

Reviewer 2 Report

Marcinkowski and colleagues present the work aiming to investigate the interaction of the FTO dioxygenase with calmodulin. They discovered this new interaction partner, mapped the interaction surface. In addition, they identified that the interaction is Ca2+ dependent. Apart from the minor comments listed below, the manuscript is well written and will be interesting to the spetialists of this field.

Minors:

Lines 98-99 – the statement „This strongly suggests that FTO is involved in calcium signaling pathways.“ is not supported by findings and too far going. Please correct.

Line 46 – explain an abbreviation XPO2

Line 56 – explain an abbreviation CREB

Line 59 – explain abbreviations NPY1R and BDNF

Line 64 – explain abbreviations p54nrb and PSPC1

Author Response

Response to Reviewer 2 Comments

Point 1: Lines 98-99 – the statement „This strongly suggests that FTO is involved in calcium signaling pathways.“ is not supported by findings and too far going. Please correct.

  • Response 1: We changed the word "strongly" to the less categorical "may".

Point 2: Line 46 – explain an abbreviation XPO2

  • Response 2: Has been corrected. Moreover, we added the “Abbreviation” section at the end of the manuscript for better clarity of the text.

Point 3: Line 56 – explain an abbreviation CREB

  • Response 3: It was corrected.

Point 4: Line 59 – explain abbreviations NPY1R and BDNF

  • Response 4: It was corrected.

Point 5: Line 64 – explain abbreviations p54nrb and PSPC1

  • Response 5: It was corrected.